# The Effectiveness of Planning Interventions for Improving Physical Activity in the General Population: A Systematic Review and Meta-Analysis of Randomized Controlled Trials

**DOI:** 10.3390/ijerph19127337

**Published:** 2022-06-15

**Authors:** Sanying Peng, Ahmad Tajuddin Othman, Fang Yuan, Jinghong Liang

**Affiliations:** 1Physical Education Department, Hohai University, Nanjing 210024, China; 2School of Educational Studies, Universiti Sains Malaysia, Penang 11800, Malaysia; judane@usm.my; 3College of International Languages and Cultures, Hohai University, Nanjing 210024, China; yuanf@hhu.edu.cn; 4Department of Maternal and Child Health, School of Public Health, Sun Yat-sen University, Guangzhou 510080, China; liangjh78@mail2.sysu.edu.cn

**Keywords:** planning interventions, action planning, coping planning, implementation intentions, physical activity, meta-analysis

## Abstract

Planning interventions such as action planning (AP) and coping planning (CP) have been recognized as influential strategies in promoting physical activity (PA), but mixed results of existing evidence have been observed. This study aims to perform a systematical meta-analysis to evaluate the efficacy of planning interventions for improving PA in the general population. Eight databases, including Medline, Embase, PsycINFO, Cochrane Library, Web of Science, ProQuest, CNKI, and Wanfang Data, were searched to locate relevant randomized controlled trials (RCTs) from their inception to 31 December 2021. In total, 41 trials with 5439 samples were included in this systematic review, and 35 trials were used in our meta-analysis. The results showed that PA was better promoted in the planned intervention group compared to the control group (SMD = 0.35, 95% CI = 0.25–0.44, I^2^ = 61.4%). Based on the subgroup analyses, we found that planning strategies were more effective among patients, males, when adopting AP intervention, when using the face-to-face sessions delivery mode, and when reinforcements were conducted during the follow-up. The findings of this study indicate that planning interventions significantly improved PA behavior, and, in some contexts, the effects performed better. Future research needs to be conducted to explore the underlying mechanisms of planning interventions and validate their effects more extensively.

## 1. Introduction

Insufficient physical activity has become a public health issue globally [1]. Regular physical activity (PA) reduces the risk of noncommunicable diseases and all-cause mortality and delivers important prevention and treatment benefits for many different physical and psychological conditions [2]. Nevertheless, according to a global survey of 1.9 million individuals in 168 countries, over one-quarter of people engage in minimum or no PA (150 min of moderate exercise per week or comparable) [3]. Globally, 81% of teenagers aged 11–17 years are insufficiently physically active [4], and older adults engage in the least amount of physical activity of all age groups [5,6]. In Canada, only 9% of children meet PA recommendations [7]. Thus, it is critical for public health practitioners to encourage regular PA by developing more effective interventions.

Despite persistent efforts to enhance physical activity (PA) through public health policies and behavior change techniques (BCTs) [8], interventions aimed at increasing public involvement in and adherence to PA have generated limited outcomes [9,10]. For instance, a comprehensive meta-analysis of 27 PA intervention studies found that the overall effectiveness of PA behavior change is d = 0.27 (95% CI = 0.17 to 0.37) [11], indicating that the effect size is small but significant [12]. More recently, Whatnall et al. [13] conducted a systematic review of 66 RCT studies that examined the effects of behavioral change interventions on step-, moderate-, vigorous-intensity PA and total PA, which identified between-group differences in only 52% of the studies. Additionally, there is currently no consensus regarding which BCTs, including web-based and mobile interventions, are more effective for promoting PA [14,15,16]. To promote PA effectively, theory-based interventions that address behavioral determinants are required.

Intention, an antecedent variable of behavior in the theory of planned behavior (TPB) [17], which has been taken as a crucial factor, plays a positive role in the domain of PA promotion. However, empirical studies have confirmed that there is still a gap between the formation of goal intention and PA behavior initiation [18]. Based on existed theories [19,20], Gollwitzer identified that implementation intentions, which are self-regulatory strategies, can help bridge the intention–behavior gap [21]. Health practitioners typically prefer the operational approaches of action planning (AP) and coping planning (CP) when applying implementation intention interventions, that is, to specify when, where, and how to perform behaviors and determine corresponding responses to obstacles [22].

Up to now, three systematic reviews have analyzed the impact of AP and CP (or implementation intention) on the initiation and maintenance of physical activity behavior [23,24,25], but these studies had some limitations regarding the inclusion of high-quality literature and outcome analyses. Bélanger-Gravel et al. [23] observed a small-to-moderate overall effect size of implementation intention in several conditions through subgroup analyses. Carraro and Gaudreau [24] conducted a meta-analysis of 23 correlational and 21 experimental studies for both spontaneous and experimentally induced interventions in the form of AP and CP. This review also confirmed small-to-moderate effect sizes for planning interventions. Neither of these reviews used data entirely from RCTs as the gold standard, nor did they probe deeply into the sources of heterogeneity among studies. A recent meta-analysis [25] found significant effectiveness only in the reinforcement condition by analyzing 13 RCTs; however, its findings were limited by the number and quality of the included studies.

Therefore, the main purpose of this study is to collect relevant, high-quality evidence to investigate the effectiveness of planning interventions in promoting physical activity using the method of systematic review and meta-analysis. The second purpose is to explore the sources of heterogeneity across studies through subgroup analyses and to analyze the differences in effect sizes of planning interventions across different interest variables.

## 2. Methods

This study was conducted according to the Preferred Reporting Items for Systematic Reviews and Meta-Analyses (PRISMA) statement [26] and Cochrane Collaboration Handbook recommendations [27].

### 2.1. Search Strategy

A comprehensive literature search that has no language and publication time constraints was conducted to identify the relevant randomized controlled trials (RCTs) that probe the effectiveness of planning or implementation intention interventions on PA in the following academic databases: Medline (via PubMed), Embase, PsycINFO, Cochrane Library, Web of Science, ProQuest, Chinese National Knowledge Infrastructure (CNKI), Wanfang Data, from their inception to 31 December 2021.

An exhaustive search was performed utilizing the medical subject headings (MeSH) combined with free text terms by employing Boolean logical operators, with the terms “Physical Activity”, “Exercise”, “Planning”, “Implementation Intentions”, “Action Planning”, “Coping Planning”, and “Randomized Controlled Trial” taken into consideration. Furthermore, as a supplementary search, we conducted a series of recursive screenings of top journals (top international journals: Health Psychology Review, Annals of Behavioral Medicine; top China journals: Sport Science, Advances in Psychological Science), grey literature, well-known publishers, and significant international academic proceedings to reduce the damage caused by the exclusion of suitable items that match our inclusion criterion. After a manual screening of the selected articles, supplementary searches were performed for other articles by important authors. In the supplement search techniques, details from all databases of search methods were displayed.

All records originally retrieved were imported into EndNote 20 (Thomson ISI Research Soft, Philadelphia, PA, USA), confirmed, and managed by two authors (S.Y.P. and F.Y.) independently and concurrently. Disagreements in this process were resolved through discussion in a meeting with the other authors (A.T.O. and J.H.L.). These steps ensure the completeness and accuracy of this study.

### 2.2. Eligibility Criteria

The following PICOS criteria were used to include relevant studies.

#### 2.2.1. Population

The participants included in this study were not restricted by age, gender, health status, region, or nationality. Given that they have freedom of movement and can participate in the physical activity promotion program set up by the researchers, they are all included in the scope of this study.

#### 2.2.2. Interventions

Acceptable treatments included planning interventions (AP or the combination of AP and CP), i.e., the core components of planning strategies, such as when, where, and how a goal action is to be performed or added and how to deal with the barriers to goal pursuing. Prerequisites such as an introduction to recommended amounts of PA or education on the benefits of PA were permitted in the intervention. Interventions that included other psychological treatments were excluded.

#### 2.2.3. Comparators

Studies were included if the control group was set up as a waiting group with no treatments or if the control group only received some PA recommendations or education about the benefits of PA. Studies with control groups other than those were excluded.

#### 2.2.4. Outcomes

Interventions that reported PA measures (e.g., number of steps, frequency of PA participation, amount of light, moderate- or high-intensity PA in time units) were included. Outcomes can be measured by any measurement instrument, including subjective self-report questionnaires, self-administered items, and objective instruments. Outcomes must include baseline and post-test measurement data.

#### 2.2.5. Study Design

Only published paralleled-group RCTs were included, including pilot RCTs, with no language limitations.

Duplications were deleted first, as indicated in our included and excluded criteria. The studies were independently chosen by two authors by examining their titles and abstracts. Following that, full-text reviewing was carried out to locate possibly suitable studies. Inconsistencies that arose in this section were avoided.

### 2.3. Data Abstraction and Risk of Bias Assessment

According to the Cochrane Consumers and Communication Review Group’s data extraction template [27], two authors performed an independent double-blind method to investigate and extract the key data of the included studies, as follows: the first author, publication year, participants, sample sizes of different groups and total of all groups, gender, age, intervention details (strategies, duration, delivery mode, and reinforcement or not), instrument and various outcomes, etc. Missing data were obtained by emailing the corresponding author of the related study or retrieving other systematic reviews that included the studies. Based on the Cochrane Risk of Bias tool [27], seven items were structured to assess the risk of bias for the included studies: (1) random sequence generation: whether the random sequence method and process are described in detail; (2) allocation concealment: whether to allocate participants strictly according to the results of random numbers; (3) blinding of participants: whether subjects and investigators were blinded; (4) blinding of outcome assessment: whether the evaluators were blinded; (5) incomplete outcome data: whether the drop-out rate was controlled within 10%; (6) selective data reporting: whether only favorable results were selectively reported; (7) other undefined biases: bias such as conflict of interest, small sample size, and baseline imbalance. Each of the items was graded as high risk of bias, low risk of bias, and unknown risk of bias, respectively. The study was assessed as low risk if more than four items were low risk. If most of the study was unknown, the study was assessed as unknown. When more than one item was high risk and less than four other items were rated as low risk, the study was considered high risk. Two reviewers independently assessed the bias risk of the included literature, and disagreements were judged by a third reviewer.

### 2.4. Statistical Analyses

According to Cochrane Collaboration Handbook recommendations [27], the present meta-analysis was conducted by a conventional pair of crossed trials for each comparison at post-intervention. To begin, the omnibus homogeneity test (Q) and I^2^ statistics were used to assess study heterogeneity, and *p* < 0.1 for Q was regarded as statistically significant, while I^2^ statistics had values of 25%, 50%, and 75%, indicating mild, moderate, and high heterogeneity [28]. Second, means and standard deviations of outcomes reported post-intervention were uniformly extracted as numerical variables for meta-analysis to ensure more accurate analysis outcomes. Moreover, to evaluate the primary effect of planning interventions, the effect sizes were pooled using the inverse variance statistical approach with random effect models. The pooled effect sizes were provided as standardized mean differences (SMDs) with respective 95% confidence intervals (CIs). SMDs were calculated by the mean and standard deviation of each comparison group. When standard errors and confidence intervals were reported instead of standard deviations, they were transformed according to statistical methods. Third, comparison-adjusted funnel plots were drawn to visually detect various forms of potential publication bias. Egger’s test was used as a supplement to the quantitative evaluation of the funnel plot to test the significance level [29]. Then, a sensitivity analysis was performed, removing the studies deemed to be at high risk of bias. Lastly, given that the role of moderator variables may be the source of heterogeneity between studies, subgroup analyses were performed to ensure the stability of the overall effect sizes. The subgroup analysis performed in this study is shown as follows: Intervention Strategy (AP vs. AP + CP); Duration (≥5 weeks vs. <5 weeks); Publication Year (year ≥ 2012 vs. year < 2012). Delivery Mode (Sessions vs. Online vs. Sessions and Online); Reinforcement or not (Yes vs. No); Participants (Patients vs. Healthy Populations); Students or not (Yes or No); Female-to-Male Ratio (≥1 vs. <1); Sample Size (≥100 vs. <100); Instrument (Objective vs. Self-report vs. Items). The above analyses were carried out in STATA 14.1 (StataCorp, College Station, TX, USA).

## 3. Results

### 3.1. Literature Selection

The initial database search yielded a total of 1790 records. Before screening, 374 duplicate records and 794 records with only simple protocols were removed. After the first round of careful screening of titles and abstracts, 254 search records entered the next step for re-examination. After 131 records were excluded, 123 studies were left for full-text review. As a result, 41 articles [30,31,32,33,34,35,36,37,38,39,40,41,42,43,44,45,46,47,48,49,50,51,52,53,54,55,56,57,58,59,60,61,62,63,64,65,66,67,68,69,70] were included in this systematic review, and only six articles [38,40,63,65,68,70] were not suitable for quantitative meta-analysis. The selection process is shown in Figure 1.

### 3.2. Characteristics of Studies

All 41 included studies were conducted in different regions or countries (23 studies from Europe [30,31,33,35,36,37,41,42,44,45,46,47,49,50,51,52,53,54,55,60,61,67,69], 7 studies from North American [32,34,43,48,56,58,65], 3 studies from South America [57,59,68], 3 studies from Asia [39,62,70], 2 studies from Australia [64,66], and 3 studies that did not report this condition [38,40,63]), with publication years ranging from 2002 to 2021. All studies were published in English except for one study published in Chinese [62] and another in Spanish [59]. In total, 2936 participants were randomized to the planning interventions group, while 2948 participants were assigned to the controlled group, with the mean age ranging between 8.06 and 73.30 years old. There were 5 studies targeting women [32,40,43,45,58] and 28 studies with more than 50% female participants [30,31,32,38,39,40,41,42,43,44,45,48,49,50,51,52,53,54,58,59,60,61,65,66,67,68,69,70]. Table 1 provides a detailed introduction to these demographic characteristics.

As regards the type of planning strategies chosen by the PA promotion intervention, the number of studies that chose AP (*n* = 21) as an intervention strategy and the number of studies that chose AP combined with CP (*n* = 20) as an intervention strategy were very similar.

The delivery of planning interventions was characterized by three typical modes. The first mode (*n* = 24) is to implement the intervention through face-to-face individual or group sessions; the second (*n* = 5) is to conduct online delivery modes, such as phone calls, emails, postal mail, pedometers, phone text messages, APP tracking, and website feedback; and the third mode (*n* = 12) is to combine face-to-face sessions with online delivery.

The duration of the intervention (from baseline to the last endpoint) of the included studies ranged from 1 week to 14 months. After the baseline interventions were applied, some studies reinforced the effects of the baseline interventions through telephone surveys, text message reminders, diary records, and face-to-face sessions. Across all studies, 20 studies used standardized self-report questionnaires tested for reliability and the validity of previous studies to assess physical activity, 15 studies employed measurement items (three of which were validated), 5 studies used objective instruments (two accelerometers and two pedometers), 1 study used diaries, and 1 study used checklists.

In most of the studies, the control groups received some motivational education, including the benefits of physical activity and WHO physical activity recommendations through face-to-face sessions, leaflets, emails, or text messages. This motivational education was also implemented simultaneously in the intervention group. Moreover, intervention completers were considered for statistical analysis in most studies and only eleven studies employed an intention-to-treat approach [36,42,45,46,47,51,52,58,60,64,69].

### 3.3. Quality of Included Studies

In 34 of the included trials [31,32,33,34,35,36,37,38,39,40,41,42,43,44,45,47,49,50,51,52,54,55,56,57,58,59,61,62,64,65,67,68,69,70], the risk of bias was classified as low or uncertain, whereas 7 studies [30,46,48,53,60,63,66] were classified with high-risk bias. In all 41 studies, sufficient random sequence generation was observed, whereas few of them had conducted their allocation concealment. Three studies [49,55,64] explicitly mentioned a sufficient blinding process of participants and researchers, whereas others were unclear. Relatively complete outcome analyses and reports were shown in most of the studies, except in nine studies [30,46,48,52,53,54,55,60,63] with relatively high drop-out rates. Regarding other bias factors, three studies [50,59,66] were deemed to have a high-risk bias. Figure 2 and Figure 3 show details on overall and individual quality.

### 3.4. Primary Outcome

Overall effect sizes were combined for the 35 trials included in the meta-analysis; 6 studies [38,40,63,65,68,70] were excluded due to unavailable data. All studies used the measurement of PA as the outcome indicator. By means of the random effect model, the planning interventions group yielded a small-to-medium significant pooled effect size (SMD = 0.35, 95% CI: 0.25, 0.44, I^2^ = 61.4%) compared to the controlled group (shown in Table 2). The effect sizes varied between −0.12 and 0.94 across the studies. A funnel plot indicated that there was no publication bias (shown in Appendix A), but the quantitative Egger test did not reveal the same result (*p*-value = 0.001) (shown in Appendix A).

### 3.5. Subgroup Analyses

The results of the predefined subgroup analyses, separated into ten subgroups, are shown in Table 2. Some subgroup analyses produced consistent findings, indicating that the items inside the subgroup were statistically significant: the duration group (≥5 W, SMD = 0.36, 95% CI: 0.23 to 0.49 vs. <5 W, SMD = 0.31, 95% CI: 0.17 to 0.46); the students group (students, SMD = 0.54, 95% CI: 0.23 to 0.85 vs. no students, SMD = 0.33, 95% CI: 0.23 to 0.0.43); however, when the delivery mode were taken into account, face-to-face sessions (SMD = 0.41, 95% CI: 0.27 to 0.55) showed a distinguished improvement compared with CG while online sessions (SMD = 0.14, 95% CI: −0.02 to 0.31) had relatively little improvement. A similar result also occurred in the other six subgroups analyses, i.e., the intervention strategy group (AP, SMD = 0.41, 95% CI: 0.24 to 0.44 vs. AP + CP, SMD = 0.30, 95% CI: 0.19 to 0.42); the publication year group (≥2012, SMD = 0.29, 95% CI: 0.16 to 0.42 vs. <2012, SMD = 0.38, 95% CI: 0.24 to 0.51); those involved in the sample size group (≥100, SMD = 0.27, 95% CI: 0.17 to 0.38 vs. <100, SMD = 0.55, 95% CI: 0.38 to 0.71); the female-to-male ratio group (≥1, SMD = 0.30, 95% CI: 0.19 to 0.42 vs. <1, SMD = 0.43, 95% CI: 0.27 to 0.58); the reinforcement group (reinforcement, SMD = 0.41, 95% CI: 0.28 to 0.53 vs. no reinforcement, SMD = 0.28, 95% CI: 0.15 to 0.42). In terms of the participants group, the magnitude of improvement in patients (SMD = 0.45, 95% CI: 0.29 to 0.61) was a little more than that of the healthy population (SMD = 0.29, 95% CI: 0.18 to 0.40). Finally, it is worth noting that the validated self-report instrument (SMD = 0.37, 95% CI: 0.25 to 0.49) had an approximately equal effect size as the non-validated one (SMD = 0.36, 95% CI: 0.20 to 0.54), while the pooled effect size of objective instruments (SMD = 0.08, 95% CI: −0.39 to 0.54) was the smallest and not statistically significant.

## 4. Discussion

Thirty-five high-quality RCTs were included in the present study for meta-analysis, and the results found that the planning strategies intervention significantly promoted physical activity in the general population, with the overall effect size (SMD = 0.35, 95% CI: 0.25, 0.44) being “small-to-medium” according to Cohen’s classification criteria of effect size (Cohen, 1988). Subgroup analyses were conducted and revealed that the planning interventions were more effective in the patient group, the group with fewer females. Moreover, the delivery mode of individual or group face-to-face sessions during the imposition of the intervention and the group that underwent post-intervention reinforcement performed better. We also found that the effects of different measurement instruments and sample sizes on the pooled effect sizes suggested that they may be sources of heterogeneity between studies.

A positive and significant intervention effect, as revealed by this study, identified that planning strategies can improve PA successfully. Bélanger-Gravel et al. [23] conducted the first meta-analysis of an AP-induced trial, showing that the planning intervention had a significant effect on physical activity, both post-intervention and at follow-up. Almost simultaneously, Carraro and Gaudreau [24] also conducted a meta-analysis combining data from both correlational and experimental studies; it showed that both spontaneous and experimentally induced AP and CP were successful in promoting physical activity. A recent meta-analysis of BCT interventions incorporating AP, conducted by Howlett et al. [16], also showed a significant small-to-moderate effect size effect of BCTs on initiating PA behaviors. This study, the largest meta-analysis of high-quality RCTs to date, further validated the significant effect of planning interventions to promote physical activity, which results from the key role that planning strategies play in behavioral change as self-regulatory strategies [71]. According to the health action process approach (HAPA), two types of planning strategies, AP and CP, play a key role in the initiation and maintenance of intended behavior [72]. The function of AP is to enhance awareness in the individual about potential future scenarios in which the behavior may be performed by making clear when, when, and how the individual would initiate the behavior [73]. CP focuses on the anticipation of barriers that may interfere with a desired activity and how to choose alternative behaviors that may be implemented to overcome those barriers [74,75]. As mental simulations of a series of behavioral processes, planning strategies facilitate the successful translation of good intentions into action through the pre-construction of situations that initiate behavior and the management of possible anticipated obstacles [76,77]. Moreover, AP and CP have been designed in HAPA as mediating variables between intentions and behaviors, helping to bridge the gap between intention and behavior [78]. Some studies have empirically confirmed that AP and CP can also moderate the relationship between intention and behavior [79,80,81,82]. From the above analysis, it can be identified that AP and CP are crucial psychological determinants of PA initiation, and future research should explore the deeper mechanisms of action of AP and CP in promoting PA [83].

The exploratory subgroup analyses conducted in this study revealed that the effects of the planning interventions differed across conditions and contexts, which contributes to a cautious interpretation of the overall effect sizes. In the intervention strategy group, the intervention effects of AP were superior to AP combined with CP, which may stem from the fact that the AP conducted in the trials was more acceptable to the participants, while the combined strategy added an extra CP to the psychological process of behavior change using an if–then format “cue” in response to behavior obstacles [24], which may have led to a decline of intervention effects. However, the combination of AP and CP is also a promising choice of an effective strategy for increasing PA, and its efficacy needs to be verified by more RCTs. Moreover, in terms of intervention delivery modes, face-to-face sessions were the most effective, with online sessions alone (e.g., telephone calls, emails, or visiting websites) being the least effective; post-intervention groups with reinforcement achieved better results. As self-regulatory strategies, planning strategies need to control the details of the interventions to be effective in promoting complex behaviors such as physical activity, so the interventions were more effective in the cases using the delivery mode of face-to-face sessions that were adept in focusing the participants’ attention and the addition of reinforcements during the follow-up period. Subgroup analysis by publication year showed a higher effect size for studies published before 2012 than those published after 2012, indicating a decreasing trend in overall effect sizes for studies in the last decade; this trend needed to be verified by more evidence. Furthermore, in the subgroup analysis of the different samples, it was found that the planned interventions were more effective with the patients than with the healthy population, which supports the idea that planning interventions were important interventions for rehabilitating patients [84]. In addition, the interventions were less effective in the population with a high proportion of females, which may be more related to the intention status of the study sample. The planning interventions had a better effect among those with PA intention [85,86], while most females are usually unintentional due to a lack of interest in PA. Although no visible difference was observed between the student and non-student groups, planning strategies remain promising interventions to promote students participating in PA because of their low cost and ease of implementation in campus settings.

Of note, the results from the instrument group in the subgroup analyses suggest that differences between instruments may have contributed to the heterogeneity of the studies. Future studies should employ validated instruments of PA, such as objective instruments (e.g., accelerometers and pedometers) or widely recognized self-report questionnaires (e.g., LTEQ [87] and IPAQ [88]). Given that objective instruments and self-reported questionnaires measure the different parts of PA and that such measurement outcomes are not equivalent [89], further investigation of more appropriate approaches to merging objective instruments and self-reported questionnaires would contribute to improving the validity of evidence based on PA measurements.

The present study is the first meta-analysis of planning interventions for PA that uses RCTs and includes a significant amount of literature covering a wide range of populations (mean age from 8.06 to 73.30 years old). Although only 11 studies used intention-to-treat analysis as a method of calculating outcome indicators, the inclusion of far more than 15 trials gives credibility to the outcomes [90]. Based on the high-quality literature included and the rigorous research procedure, the findings of this study elucidate the broad effectiveness of planning interventions. As low-cost interventions that can be delivered in a variety of ways, planning interventions can be easily disseminated and promoted to a wide range of populations, providing them with promising strategies used in the public health domain to increase physical activity and prevent noncommunicable diseases caused by sedentary and physical inactivity.

Several uncontrollable limitations also affected the results of this study. Firstly, most of the included trials in this study were completed in developed countries, and they fail to reflect the actual characteristics of the broader sample. In addition, few trials completed the registration process on the relevant trial platforms, which may directly affect the stability of our evidence. Finally, although we used quantitative analyses to ensure the accuracy of the effects of planning interventions in promoting PA, the sources of moderate heterogeneity observed (e.g., different planning intervention strategies, intervention delivery models, samples, and sample size selection, etc.) need to be further explored.

## 5. Conclusions

This review identifies that planning interventions are effective in improving PA behavior among the general population. In addition, the results of this review provide sufficient evidence that the effects of planning interventions vary according to different moderators and contexts. As effective intervention strategies with low cost, planning intervention should be broadly promoted and applied by health practitioners and policymakers.

## Figures and Tables

**Figure 1 ijerph-19-07337-f001:**
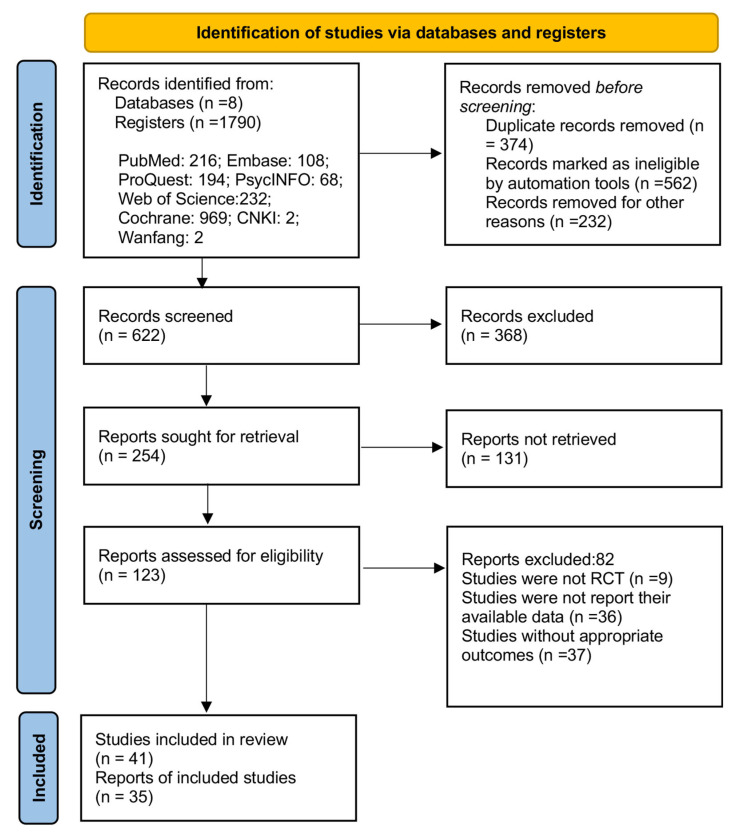
Flow chart of study design by PRISMA 2020.

**Figure 2 ijerph-19-07337-f002:**
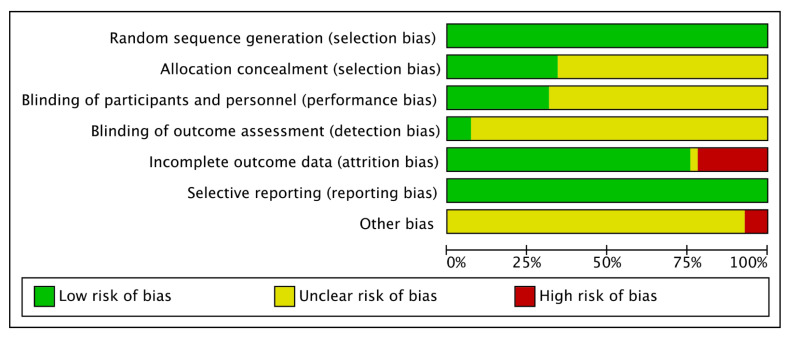
Risk of bias graph.

**Figure 3 ijerph-19-07337-f003:**
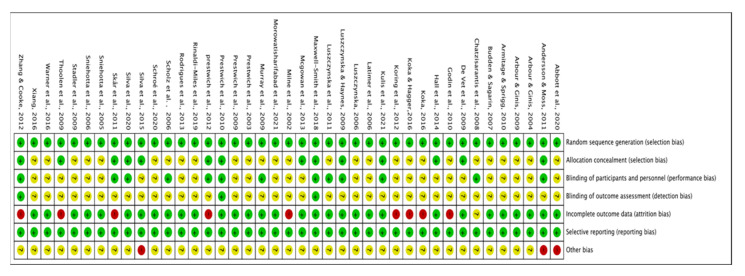
Risk of bias summary for included studies [30,31,32,33,34,35,36,37,38,39,40,41,42,43,44,45,46,47,48,49,50,51,52,53,54,55,56,57,58,59,60,61,62,63,64,65,66,67,68,69,70].

**Table 1 ijerph-19-07337-t001:** Characteristics of studies.

Publication	Sample Size		Female (%)	Age		Country	Participant	Intervention Strategy	Delivery Mode	Duration	Reinforcement	Instrument	Outcome
	**IG**	**CG**		**IG**	**CG**								
Kulis et al., 2021	82	76	64.40	43.86 ± 17.02		Poland	Inactive adults	AP + CP	Session and online	36 W	3 sessions + 4 Phone	Accelerometer	No. MVPA min/day
Schroé et al., 2020	38	46	72.80	35.66 ± 15.83	33.33 ± 16.93	Dutch	General adults	AP + CP	Session and online	5 W	every week by email	IPAQ	No. MVPA min/week
Maxwell-Smith et al., 2018	34	34	50.00	65.26 ± 7.41	62.88 ± 8.37	Australia	Cardiovascular risk survivors	AP + CP	Session and online	12 W	3 phones	Accelerometer	No. MVPA min/week
Abbott et al., 2020	14	13	63.33	37.7 ± 13.4	48.90 ± 14.50	Australia	Inactive adults	AP	Session and online	12 W	weekly via online	IPAQ	No. MVPA min/week
Koka, 2016	54	64	50.63	14.79 ± 0.71		Estonia	Adolescents	AP	Session	1 M	No	Items (not validated)	No. times of LTPA > 30 min/week
Milne et al., 2002	79	93	73.00	20.04 ± 2.23		UK	Undergraduate students	AP	Session	2 W	No	Items (not validated)	No. times MVPA > 20 min/week
Arbour and Ginis, 2004	24	19	100	45.38 ± 7.55	47.78 ± 7.03	Canada	Sedentary adults	AP	Session	8 W	No	Diary (not validate)	No. times PA at the recommended level/week
Latimer et al., 2006	19	18	43.24	40.89 ± 11.56	40.94 ± 10.85	North America	Spinal Cord Injury patients	AP	Session	8 W	2 emails	PARA–SCI	Physical activity duration (min/day)
Luszczynska, 2006	59	55	36.00	54:25 ± 6.85		Poland	Myocardial infarction patients	AP	Session	8 W	No	Item (not validated)	Scores expressing frequency
Prestwich et al., 2009	29	34	58.06	23.76 ± 4.64		UK	Inactive undergraduate students	AP	Session and online	4 W	phone	Item (not validated)	No. times of MVPA > 30 min/week
Stadler et al., 2009	133	133	100	41.33 ± 5.91	41.22 ± 6.48	Germany	General adults	AP + CP	Session	16 W	4 sessions	BTDPAR	No. MVPA min/week
Armitage and Sprigg, 2010	39	38	49.35	8.06 ± 1.63		UK	Children	AP	Session	6 W	2 sessions	Items (validated)	Scores expressing frequency
Prestwich et al., 2010	40	46	63.76	22.19 ± 5.01	23.62 ± 4.49	UK	Inactive adults	AP	Session and online	4 W	3 text messages by mobile phone	Items (validated)	No. days exercised for 30 min/week
Andersson and Moss, 2011	13	14	78.03	27.00 ± 6.80	26.20 ± 6.70	UK	Inactive adults	AP	Online	2 W	No	LTEQ	MVPA Occasions/week
Luszczynska et al., 2011	36	22	56.90	48.17 ± 17.89		Poland	Diabetes patients	AP	Session	4 W	No	One Item	Scores expressing frequency
Koring et al., 2012	445	438	67.95	42.92 ± 14.91	43.86 ± 13.66	Germany	General adults	AP + CP	Online	3 W	No	IPAQ	MVPA min/week
Zhang and Cooke, 2012	22	21	48.81	20.56 ± 1.62		UK	Undergraduate students	AP + CP	Online	4 W	No	Items (not validated)	No. times MVPA > 20 min/week
Mcgowan et al., 2013	141	141	0	68.40	67.90	Canada	Prostate cancer survivors	AP + CP	Session	1 M	No	The index of LTEQ	No. MVPA min/week
Rodrigues et al., 2013	69	67	36.00	56.70 ± 9.10		Brazil	Coronary heart disease patients	AP + CP	Session and online	2 M	4 telephones	Baecke-HPA	No. times walked at least 30 min last month
Hall et al., 2014	24	28	100	73.30 ± 7.17	73.11 ± 6.66	Canada	Older Adult Women	AP + CP	Session and online	4 W	4 telephones	Stanford 7-day Recall	No. times of half-hour VPA/week
Silva et al., 2015	15	15	66.67	61.27 ± 6.26	59.87 ± 12.61	Brazil	Type II diabetics patients	AP + CP	Session and online	2 M	Telephone	IPAQ	No. MVPA min/week
Sniehotta et al., 2005	65	79	18.50	57.70 ± 10.30		Germany	Cardiac rehabilitation patients	AP + CP	Session and online	4 M	Diary	Adapted version of KPAS	No. general exercise min/week
Sniehotta et al., 2006	62	81	22.00	59.30 ± 10.00		Germany	Cardiac rehabilitation patients	AP + CP	Session	2 M	No	Items (not validated)	No. all activity min/week
Murray et al., 2009	29	23	100	30.50 ± 9.80		Canada	General adults	AP	Session	11 W	3 times repetition	checklist at the gym (not validated)	No. sessions/week
Thoolen et al., 2009	119	108	40.00	62.00 ± 4.90	61.90 ± 5.60	Dutch	Diabetes patients	AP + CP	Session	12 M	4 sessions	PASE	Scores expressing amount
Prestwich et al., 2003	18	18	51.20	21.31 ± 4.39		UK	General adults	AP	Session	4 W	No	Items not validated	No. sessions/week
Xiang, 2016	31	32	46.03	10. 25 ±0. 43		China	Elementary school students	AP	Session	1 M	Physical education course	Physical Activity Questionnaire for Children	Scores expressing frequency
Godin et al., 2010	108	113	61.60	38.20 ± 10.20	37.10 ± 11.00	Canada	General adults	AP	Online	6 M	No	Items (not validated)	Scores expressing frequency
Scholz et al., 2006	103	95	17.70	58.50 ± 10.60		Germany	Cardiac rehabilitation patients	AP + CP	Session and online	12 W	Diary	Adapted version of the IPAQ	No. MVPA min/week
Luszczynska and Haynes, 2009	104	78	89.00	28.73 ± 9.51		UK	General adults	AP + CP	Session	4 M	Repeat 3 times	Items (not validated)	Score expressing frequency
Skår et al., 2011	335	315	63.40	22.80 ± 6.70		UK	University students	AP + CP	Online	6 W	No	Items (validated)	Scores expressing frequency
De Vet et al., 2009	172	206	67.00	45.90 ± 10.34		Dutch	General adults	AP	Session	6 M	No	SQUASH	No. all activity min/week
Chatzisarantis et al., 2008	92	35	72.44	20.71 ± 6.95		Singapore	Sedentary students	AP	Session	5 W	No	LTEQ	Scores expressing frequency
Prestwich et al., 2012	45	57	79.44	42.33 ± 10.62	41.55 ± 10.71	UK	General adults	AP	Session	6 M	No	SWET	Scores expressing frequency
Warner et al., 2016	25	67	75.20	70.34 ± 4.89		Germany	General adults	AP + CP	Session	14 M	No	the index of the validated PRISCUS-PAQ	No. overall PA min/week
Koka and Hagger, 2016	62	72	NR	14–15		NR	High-school students	AP	Session	3 M	No	Items (not validated)	Times of MVPA > 30 min/week
Arbour and Ginis 2009	35	32	100	48.17 ± 9.61		NR	Sedentary women	AP	Session	11 W	Record daily steps	Pedometer	Steps/day
Budden, 2007	NR	NR	60.00	NR	NR	NR	General adults	AP	Session	1 W	No	Items (not validated)	Scores expressing frequency and duration
Morowatisharifabad et al., 2021	63	62	77.60	25–65		Iran	Type II diabetics patients	AP + CP	Session	3 M	9 sessions	IPAQ	METs level of PA/week
Silva et al., 2020	33	32	67.69	60.21 ± 10.83	63.25 ± 10.33	Brazil	Type II diabetics patients	AP + CP	Session	12 M	3 on-site sessions	GSLTPAQ	Scores expressing frequency
Rinaldi-Miles et al., 2019	26	28	87.00	47.70 ± 9.019		USA	Inactive adults	AP + CP	Session and online	8 W	N	Pedometer	Steps/day

Notes: AP: action planning; Baecke-HPA: Baecke Questionnaire of Habitual Physical Activity; BTDPAR: Bouchard Three-Day Physical Activity Record; CG: controlled group; CP: coping planning; GSLTPAQ: Godin–Shephard Leisure-Time Physical Activity Questionnaire. IG: intervention group; IPAQ: International Physical Activity Questionnaire; KPAS: Kaiser Physical Activity Survey; LTEQ: Leisure Time-Exercise Questionnaire; LTPA: leisure-time physical activity; M: month; METs: metabolic equivalents; MVPA: moderate–vigorous physical activity; PARA–SCI: Physical Activity Recall Assessment for Individuals with Spinal Cord Injury patients; PASE: Physical Activity Scale for the Elderly; PRISCUS-PAQ: PRISCUS-Physical Activity Questionnaire; SQUASH: Dutch Short Questionnaire to Assess Health Enhancing Physical Activity; SWET: self-report walking and exercise tables; W: week.

**Table 2 ijerph-19-07337-t002:** Primary results and subgroup analyses.

Moderator	Category	Heterogeneity Test			SMD and 95% CI	Double-Tails Test		Studies	Sample Size
		x^2^	*p*	I^2^ (%)		Z	*p*		
Intervention strategy	Action Planning	45.54	<0.001	62.7	0.41 (0.24, 0.44)	4.80	<0.001	18	1801
	Action Planning and Coping Planning	41.87	<0.001	61.8	0.30 (0.19, 0.42)	5.00	<0.001	17	3638
	Overall	88.06	<0.001	61.4	0.35 (0.25, 0.44)	7.02	<0.001	35	5439
	Between	1.03	0.310						
Publication Year	≥2012	20.92	0.074	37.9	0.29 (0.16, 0.42)	4.42	<0.001	14	2138
	<2012	66.95	<0.001	70.1	0.38 (0.24, 0.51)	5.36	<0.001	21	3301
	Overall	88.06	<0.001	61.4	0.35 (0.25, 0.44)	7.02	<0.001	35	5439
	Between	0.80	0.371						
Duration	≥5W	70.00	<0.001	68.6	0.36 (0.23, 0.49)	5.49	<0.001	23	3556
	<5W	18.05	0.080	39.1	0.31 (0.17, 0.46)	4.31	<0.001	12	1883
	Overall	88.06	<0.001	61.4	0.35 (0.25, 0.44)	7.02	<0.001	35	5439
	Between	0.24	0.628						
Delivery Mode	Sessions	47.54	<0.001	62.1	0.41 (0.27, 0.55)	5.85	<0.001	19	2569
	Online	8.27	0.082	51.6	0.14 (−0.02, 0.31)	1.69	0.090	5	1824
	Sessions and online	20.66	0.024	51.6	0.34 (0.15, 0.53)	3.59	<0.001	11	1046
	Overall	88.06	<0.001	61.4	0.35 (0.25, 0.44)	7.02	<0.001	35	5439
	Between	6.20	0.045						
Reinforcement	Yes	28.24	0.042	39.8	0.41 (0.28, 0.53)	6.48	<0.001	17	1950
	No	48.48	<0.001	67.0	0.28 (0.15, 0.42)	4.12	<0.001	18	3489
	Overall	88.06	<0.001	61.4	0.35 (0.25, 0.44)	7.02	<0.001	35	5439
	Between	1.74	0.187						
Participants	Patients	20.52	0.025	51.3	0.45 (0.29, 0.61)	5.54	<0.001	11	1437
	Healthy population	57.19	<0.001	59.8	0.29 (0.18, 0.40)	5.01	<0.001	24	4002
	Overall	88.06	<0.001	61.4	0.35 (0.25, 0.44)	7.02	<0.001	35	5439
	Between	2.62	0.105						
Female/Male	≥1	58.17	<0.001	60.5	0.30 (0.19, 0.42)	5.14	<0.001	24	3975
	<1	20.66	0.024	51.6	0.43 (0.27, 0.58)	5.31	<0.001	11	1464
	Overall	88.06	<0.001	61.4	0.35 (0.25, 0.44)	7.02	<0.001	35	5439
	Between	1.54	0.215						
Students	Yes	25.77	<0.001	72.8	0.35 (0.24, 0.45)	2.68	0.007	8	1313
	No	60.91	0.001	57.3	0.34 (0.09, 0.59)	6.44	<0.001	27	4126
	Overall	88.06	<0.001	61.4	0.35 (0.25, 0.44)	7.02	<0.001	35	5439
	Between	0.00	0.951						
Sample Size	≥100	58.90	<0.001	66.0	0.27 (0.17, 0.38)	5.09	<0.001	21	4734
	<100	14.59	0.334	0.0	0.55 (0.38, 0.71)	6.53	<0.001	14	705
	Overall	88.06	<0.001	61.4	0.35 (0.25, 0.44)	7.02	<0.001	35	5439
	Between	7.45	0.006						
Instrument	Objective	2.69	0.101	62.9	0.08 (−0.39, 0.54)	0.32	0.749	2	226
	Self-report(validated)	63.95	<0.001	67.2	0.37 (0.25, 0.49)	5.99	<0.001	22	4189
	Self-report(no)	15.31	0.121	34.7	0.36 (0.20, 0.52)	4.34	<0.001	11	1024
	Overall	88.06	<0.001	61.4	0.35 (0.25, 0.44)	7.02	<0.001	35	5439
	Between	1.46	0.481						

Notes: CI: confidence interval; SMD: standard mean differences.

## Data Availability

Data generated or analyzed during this study are included in this published article or in the data repositories listed in the references.

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
