# Peer review of "The Effectiveness of Planning Interventions for Improving Physical Activity in the General Population: A Systematic Review and Meta-Analysis of Randomized Controlled Trials"

_ijerph, 2022, doi:10.3390/ijerph19127337_

Round 1
Reviewer 1 Report
Thank you for providing me the opportunity to review this well-written manuscript. Minor concerns are outlined below.
METHOD
The full sentence "...No ethical approval or participant consent..." can be suppressed as it is an obvious matter regard systematic reviews.
I recommend address 2nd, 3rd, and 4th paragraphs of the methods in a specific subtopic called Search Strategy or Information Sources.
Why the authors did not decide to resolve disagreements only by discussion (consensus)? Decision by the first author sounds like an arbitrary decision.
In the PICOS strategy the authors did not mention the health status of the participants, but it is mandatory.
In the statistical approach, the authors mentioned "In the case of studies where baseline data were not available, but there 174 were no significant differences between groups mentioned in analysis, the post-experi-175 mental mean and standard deviation were used." This approach may create a bias in the results, since values delta (differences between pre- and post-intervention) is fairly lesser than the post-intervention mean isolated. For instance, if mean steps of pre-intervention and post-intervention are 5000 and 8500, respectively, the difference is 3500 steps. Imagine comparing 3500 steps (delta) against a mean value of post-intervention of other study (i.e. 9000 steps)?! I recommend just use mean and sd differences or post-intervention mean and sd.
I would like to read an explanation for the subgroup analysis regard the publication year.
DISCUSSION
I suggest an approach in the discussion section about further investigations merging self-report instruments and objective instruments of PA.
Reviewer 2 Report
The introduction is very well and contains all the necessary information. PRISMA guidelines represent the gold standard in the creation of meta-analyzes. The Method section is generally well written and does not require changes. I think the authors have done an excellent job and no further changes are needed.
Author Response
Dear reviewer,
Thank you for your letter and for the reviewers’ comments concerning our manuscript entitled “The Effectiveness of Planning Interventions for Improving Physical Activity in General Population: A Systematic Review and Meta-analysis of Randomized Controlled Trails” (Manuscript ID: ijerph-1735203). Those comments are all valuable and very helpful for revising and improving our paper, as well as the important guiding significance to our research. Thank you again for your high approval of our research work, and we will further optimize the manuscript according to other reviewers’ suggestions to meet publication requirements.
Best wishes.
Your sincerely
Reviewer 3 Report
Dear authors, congratulations for this SR.
I recommend some revisions.
Results:
Line 201: "in my systematic review,", i suggest changing to "this systematic review".
Line 222: Table 1 is very difficult to read, please improve that table.
Line 267: Figure 3 are difficult to read, please improve that figure.
Line 287: Add a space between "online" and "(" : "online(SMD=0.44, 95% CI: 0.27 to 0.61)"
Round 2
Reviewer 3 Report
Dear authors,
Thank you for considering the suggestions, and for improving this systematic review.
Kind regards.
Author Response

(The authors gave the same response as above.)
